# CT Images in Follicular Lymphoma: Changes after Treatment Are Predictive of Cardiac Toxicity in Patients Treated with Anthracycline-Based or R-B Regimens

**DOI:** 10.3390/cancers16030563

**Published:** 2024-01-29

**Authors:** Fabiana Esposito, Valeria Mezzanotte, Cristiano Tesei, Alessandra Luciano, Paola Elda Gigliotti, Andrea Nunzi, Roberto Secchi, Cecilia Angeloni, Maria Pitaro, Federico Meconi, Martina Cerocchi, Francesco Garaci, Adriano Venditti, Massimiliano Postorino, Marcello Chiocchi

**Affiliations:** 1Hematology, Department of Biomedicine and Prevention, University of Rome Tor Vergata, 00133 Rome, Italy; valeria.mezzanotte@ptvonline.it (V.M.); c_tesei@yahoo.it (C.T.); andrea.nunzi@ptvonline.it (A.N.); robsecchi922@gmail.com (R.S.); adriano.venditti@uniroma2.it (A.V.); massimiliano.postorino@ptvonline.it (M.P.); 2Department of Diagnostic Imaging and Interventional Radiology, University of Rome Tor Vergata, 00133 Rome, Italy; alessandra.luciano@students.uniroma2.eu (A.L.); paolaelda.gigliotti@students.uniroma2.eu (P.E.G.); cecilia.angeloni@ptvonline.it (C.A.); maria.pitaro@ptvonline.it (M.P.); martina.cerocchi@ptvonline.it (M.C.); francesco.garaci@uniroma2.it (F.G.); marcello.chiocchi@ptvonline.it (M.C.); 3Fondazione Policlinico di Roma Tor Vergata, 00133 Rome, Italy; federico.meconi@ptvonline.it

**Keywords:** follicular lymphoma, R-CHOP, R-B, CT, epicardial adipose tissue, myocardial extracellular volume

## Abstract

**Simple Summary:**

Epicardial adipose tissue radio intensity and myocardial extracellular volume in computed tomography were investigated and evaluated at disease onset and after chemotherapy in patients with follicular lymphoma treated with R-CHOP and R-bendamustine regimens. The different radiomic changes in epicardial adipose tissue were evaluated in a cohort of patients treated with an anthracycline and steroid-containing regimen (R-CHOP) compared to a cohort of patients treated with R-bendamustine. Similarly, myocardial extracellular volume was assessed as an early imaging marker of myocardial fibrosis and thus of cardiological damage in the two different populations. This study analysis aims to detect the potential role of epicardial adipose tissue density and myocardial extracellular volume as early biomarkers of cardiotoxicity by comparing two valid chemo-immunotherapy regimens in follicular lymphoma patients.

**Abstract:**

The aim of this study is to evaluate changes in epicardial adipose tissue (EAT) and cardiac extracellular volume (ECV) in patients with follicular lymphoma (FL) treated with R-CHOP-like regimens or R-bendamustine. We included 80 patients with FL between the ages of 60 and 80 and, using computed tomography (CT) performed at onset and at the end of treatment, we assessed changes in EAT by measuring tissue density at the level of the cardiac apex, anterior interventricular sulcus and posterior interventricular sulcus of the heart. EAT is known to be associated with metabolic syndrome, increased calcium in the coronary arteries and therefore increased risk of coronary artery disease. We also evaluated changes in ECV, which can be used as an early imaging marker of cardiac fibrosis and thus myocardial damage. The R-CHOP-like regimen was associated with lower EAT values (*p* < 0.001), indicative of a less active metabolism and more adipose tissue, and an increase in ECV (*p* < 0.001). Furthermore, in patients treated with anthracyclines and steroids (R-CHOP-like) there is a greater decrease in ejection fraction (EF *p* < 0.001) than in the R-B group. EAT and ECV may represent early biomarkers of cardiological damage, and this may be considered, to our knowledge, the first study investigating radiological and cardiological parameters in patients with FL.

## 1. Introduction

Follicular lymphoma is generally an indolent disease and it is the most common non-Hodgkin lymphoma (NHL), after diffuse large B-cell lymphoma (DLBCL), in western Europe and the United States [1].

It accounts for 70% of indolent lymphomas and occurs mainly in people with a median age of 65 [2,3].

Follicular lymphoma is a heterogeneous disease, with variable prognoses based on phenotypic, genetic and clinical expression. Its course varies from indolent lymphoma, requiring no treatment but only a ‘watch and wait’ approach, to neoplasms requiring radiotherapy treatment or a first-line systemic chemo-immunotherapy approach. Treatment strategies depend on disease stages and clinical factors. Indications for treatment are rapidly growing tumors or large masses, B symptoms, impaired organ function, symptomatic disease or cytopenias [1,4,5]. The natural history of the disease and therefore its prognosis have benefited from the introduction, in recent decades, of rituximab combined with chemotherapy regimens or in maintenance therapy, improving in some patients progression-free survival (PFS) and overall survival (OS) [6,7,8].

In patients requiring first-line systemic chemo-immunotherapy, rituximab in combination with anthracycline-based chemotherapy (R-CHOP-like), such as cyclophosphamide doxorubicin, vincristine and prednisone (CHOP), or rituximab in combination with bendamustine (R-B) represents the first therapeutic choice [4,9,10].

Often, therefore, in clinical practice, the choice of treatment between the two regimens is based on the patient’s clinical characteristics such as comorbidity and cardiological risk, performance status (ECOG scale), or more simply, the patient’s choice not to undergo chemotherapy, which may cause alopecia as a side effect.

Although the use of R-CHOP is recommended in grade 3B, which behaves like DLBCL, there are no other indications of preference between the two regimens [11], with the exception if the patient needs a rapid response, in which case the use of the R-CHOP regimen is recommended and, on the other hand, if the patient has a history of heart failure, in which case an anthracycline-free regimen such as R-B is recommended [12,13].

Anthracycline (AC) cardiotoxicity has been known for decades, especially with regard to doxorubicin, with a still unclear mechanism of action and to date, there are no well-defined prevention strategies [12,13]. Cardiac anomalies can occur years after therapy and may be more frequent in the presence of risk factors such as advanced age, male gender, overweight and, above all, the cumulative dose of AC [14]. Anthracyclines have long been linked with the dose-related development of fibrosis [15,16]. Clinically, the effects may be heterogeneous and in the long term may lead to left ventricular ejection fraction (LVEF) depression or symptomatic heart failure [15,17,18]. In older FL patients, liposomal doxorubicin, characterized by lower cardiotoxicity, is often used to prevent detrimental cardiac effects in R-CHOP-like (R-COMP) chemo-immunotherapy regimes or regimens that omit the use of anthracyclines such as R-CVP [4,19].

Early markers, such as troponins, although non-specific, may give an indication of cardiac damage [12]. Nevertheless, nowadays, there are no early markers of cardiac damage and cardiac extracellular volume (ECV) calculated by CT, which is used in lymphoma patients in common clinical practice, that could be a reliable and valid marker capable of detecting both anthracycline-induced cardiac injury and patients who would not be candidates for cardiotoxic drugs at onset due to cardiological comorbidities [18]. Cardiac extracellular volume is a measure of cardiac fibrosis, reflecting the loss of myocardiocytes, and its value increases in the presence of cardiac damage. Its normal value is 25.3 ± 3.5% [20,21].

CT for the measurement of ECV appears to be as reliable as magnetic resonance imaging (MRI), the latter being more difficult to access, both because it is not routinely performed in the evaluation of disease in patients with FL and it has a high cost of implementation or execution [22,23,24,25].

Another parameter that appears to be a potential marker of changes related to anthracycline cardiotoxicity is the epicardial adipose tissue (EAT), thoracic visceral fat located on the surface of the heart between the myocardium and the visceral pericardium [26].

Increased EAT appears to have pro-inflammatory, pro-thrombotic and pro-atherogenic effects. Thus, EAT has been associated with coronary artery disease (CAD), arrhythmias, heart failure (HF) and other cardiovascular conditions [27,28,29,30]. EAT is strongly and independently correlated with increased cardiac mortality [31].

The density of EAT is expressed in Hounsfield units (HU) and a low-density EAT is characterized by a higher lipid content and, as such, a pro-atherogenic profile due to lower levels of adiponectin.

The primary endpoint of this experimental study is to evaluate changes in cardiac radiomic parameters in patients treated with anthracyclines and high-dose steroids (R-CHOP group) compared to the R-B treated group. Considering the wide use of AC in FL patients and the lack of specific biochemical markers to predict early cardiotoxicity, it is deemed interesting to assess changes in EAT and ECV measured by CT in this patient setting and their potential role in predicting chemotherapy cardiotoxicity.

As a secondary objective, this study also aims to evaluate the change in the ejection fraction (EF) measured by echocardiogram before and after chemoimmunotherapy treatment.

This research may contribute to our future efforts by offering valuable insights. The parameters identified may prove be useful in guiding therapeutic decisions and improving the assessment of potential prognostic value.

## 2. Materials and Methods

### 2.1. Study Population

Retrospectively, the study population has been selected from a database of patients diagnosed with FL referred to our institute between January 2011 and May 2019. The median ± SD follow-up of the enrolled population is 67 ± 37 months. The minimum follow-up is 48 months (6 patients). The database includes 120 patients with FL between 60 and 80 years old and 40 have been excluded due to technical problems with imaging, radiological examinations not performed in our center or because they did not satisfy the inclusion criteria.

Patients under 60 and over 80 years of age were excluded from the analysis to make the population homogeneous.

Eighty patients with newly diagnosed FL grade 1–3A according to the WHO 2016 classification were selected and treated with first-line R-CHOP-like or R-B chemotherapy regimens.

The inclusion criteria comprised an age between 60 and 80 years, first-line treatment with 6 cycles of R-CHOP (combining rituximab with cyclophosphamide, doxorubicin, vincristine and prednisone) or R-COMP (when using liposomal doxorubicin) every 21 days, or 6 cycles of R-B (including rituximab and bendamustine) every 28 days.

All patients underwent a total body CT at baseline and at the end of treatment suitable for radiomic analysis at the Department of Diagnostic Imaging Radiology at the Policlinico Tor Vergata.

Furthermore, the patients underwent an echocardiogram at the onset and end of chemo-immunotherapy treatment.

Patients diagnosed with FL grade 3B, patients with FL transformed into DLBCL and patients who had not undergone the radiological investigations or echocardiogram at our institution were excluded.

Table 1 summarizes the characteristics of the evaluated patients.

For each patient, their age, sex, histology, Ann Arbor stage, performance status (ECOG scale) and hemoglobin value were reported in the database and we calculated the Follicular Lymphoma International Prognostic Index (FLIPI). We also indicated the date of diagnosis, response to end of treatment, date of progression, if present, and date of last follow-up.

Moreover, among the descriptive parameters, from the weight and height of the patients, we calculated the body max index (BMI).

As radiomic parameters, we measured and calculated the EAT, ECV and EF and EF measured by echocardiogram at the onset and end of chemo-immunotherapy treatment.

### 2.2. Radiomics Parameters

CT images were retrospectively evaluated by a team of radiologists. Furthermore, a radiologist with 10 years of expertise in cardiovascular imaging reviewed the CT scans to obtain the radiomic parameters.

Full-body CT scans were performed with a 128-layer CT scanner (GE-Healthcare; Revolution EVO, CT, General Electrics Medical System, Milwaukee, WI, USA) in the cranial–caudal scan direction. The acquisition protocol included a baseline scan and three subsequent scans performed after administrating iodinated contrast (Iomeron 350 mg/mL, Bracco, injected volume 100–120 mL) followed by 30–50 mL of saline (injection rate of 3 mL/s).

The bolus tracking technique was used to obtain the post-contrast scans: a region of interest (ROI) was positioned in the descending aorta at the thoracic–abdominal passage and a threshold of 120 HU was set to start the scan. Three post-contrast phases were obtained: arterial phase (generally about 15–18 s after contrast injection), portal phase (70–80 s after contrast injection) and tardive phase (about 3–5 min after contrast injection).

Radiation doses were reported using the following formula: dose length product (DLP)—expressed in mGy—× cm (DLP values for each patient were extracted).

To calculate the ECV, the operator selected the optimum slice to observe the four cardiac chambers from an axial perspective.

The measurements were taken by manually inserting one ROI in the left ventricle’s ‘blood’ pool and a second ROI in the middle of the interventricular septum, avoiding papillary muscles. The same ROIs were set in the pre-contrast and post-contrast scans (portal phase). ROIs were measured in two stages (basal and portal) at time 0 and at the end of chemo-immunotherapy to provide Hounsfield units (HUs) to be used in the ECV calculation.

ECV was determined by the following equation, as previously carried out by Miller et al. [32].
ECV = (1 − haematocrit) × [(HUmyopost − HUmyopre)/(HUbloodpost − HUbloodpre)]
where HUmyopost and HUmyopre, respectively, stand for HU of septal myocardial muscle measured at portal phase and baseline scans; HUbloodpost and HUbloodpre, respectively, stand for HU of left ventricle “blood pool” measured at portal phase and baseline scans.

Standard deviations of ROIs were calculated to avoid contamination of myocardial HU measurements by motion artefacts. Altered values were not considered in this study (Figure 1).

The epicardial adipose tissue (EAT) density was assessed using a specialized workstation (GE Healthcare, General Electric, Boston, MA, USA) at baseline and at the end of therapy. EAT density was measured on basal scans in a 4-chamber projection, at the level of the anterior interventricular sulcus, at the origin of the posterior interventricular artery within the posterior interventricular sulcus and at the level of the cardiac apex (Figure 2).

### 2.3. Statistical Analysis

A preliminary analysis to select a proper sample size was performed, referring to published data [33]. A sample size of 78 was calculated, assuming a normal distribution of radiomic variables and a power of 0.90, with alpha = 0.05. In coherence with the standard statistical practices and benchmark, we decided to include two additional patients who were eligible for the study. Continuous data are expressed as mean or median with SD, whereas categorical data are expressed as percentages and proportions. The statistical significance of categorical data was analyzed using the chi-squared test, and continuous data were analyzed using Student’s t-test or the Mann–Whitney U test. Wilcoxon signed-rank tests in cases of ordinal variables and repeated measures ANOVA for the comparisons with the post hoc test (radiomic variables pre- and post-therapy), which take into account multiple tests, were used as appropriate. Delta (Δ) values (post-therapy mean value and pre-therapy mean value) and delta percentages were calculated to evaluate the change in the values of pre- and post-therapy between radiomic groups. Boxplots were used to show the epicardial fat values at the cardiac apex level, anterior intraventricular sulcus and posterior intraventricular sulcus, the mean epicardial fat, and the difference in ECV and EF between pre- and post-therapy values. Spearman’s correlation method was used to analyze the correlation between BMI and radiomic cardiac variables pre- and post-therapy.

Progression-free survival (PFS) was estimated using the Kaplan–Meier product-limit estimator. Differences in curves were assessed using log-rank tests. Univariate Cox regression analysis models were performed using stepwise selection steps to analyze the relationship between a risk factor and the incidence of a given clinical outcome, correcting for one or more confounders. Confidence intervals were calculated at the 95% level. For all tests, *p* < 0.05 was accepted as statistically significant. For all analyses, we used the R system software (R Foundation for Statistical Computing c/o Institute for Statistics and Mathematics, Wirtschaftsuniversität, 1020 Wien, Austria), Microsoft Corporation (2022) and GraphPad Prism ver. 9.0.0, G*Power (Release 3.1.9.6) Microsoft Excel for Mac. Retrieved from https://office.microsoft.com/excel (accessed on 8 November 2023).

## 3. Results

The main clinical and demographic characteristics of the study population are summarized in Table 1. Eighty patients with FL were included in the study (42 women and 38 men). The mean age was 68 years with an age range chosen between 60 and 80 years. Of the 80 patients observed, 49 were treated according to the R-CHOP scheme and 31 followed the R-bendamustine scheme.

In the R-CHOP cohort, the ratio of males to females overlaps (M/F ratio = 1), whereas in the R-B group, men are more prevalent (M/F ratio = 1.4). There were no statistically significant differences in terms of median age in the two groups at the time of treatment (66 years in the R-CHOP group vs. 70 years in the R-B group, *p* = 0.053) (Table 1).

We also calculated the FLIPI score which showed, in both groups, a higher rate in the intermediate risk category (54.8% in R-B and 58.3% in R-CHOP). Patients with low FLIPI had a lower rate (9.7% in R-B and 2.1% in R-CHOP). The pre- and post-therapy radiomic data in the two patient groups are shown in Table 2 and Figure 3. The EAT (at the level of the cardiac apex and the anterior and posterior interventricular sulcus), and mean EAT, ECV and EF were calculated in both patient cohorts.

There were no statistically significant differences in the two groups of patients in terms of pre-treatment ECV% (30.4 ± 6.0 in the R-CHOP group and 31.5 ± 6.3 in the R-B group, *p* = 0.429) and EAT HU (−238.08 ± 43.97 in the R-CHOP group vs. −227.58 ± 41.17 in the R-B group, *p* = 0.290). In each group, there was an increase in the mean values pre- and post-therapy for ECV and a decrease in EF and EAT (Figure 3). ECV and EAT parameters were calculated before and after treatment and thus also their variation. The difference in mean values pre- and post-therapy (delta or Δ) of ECV and EAT in the R-CHOP patient group is higher than in the R-B group and the results are shown in Table 3, taking into account the relevant values of the two groups (Δ ECV/R-CHOP = 12.9%, Δ ECV/R-B = 4.7%, *p* < 0.001; Δ EAT/R-CHOP = −33.94 HU, Δ EAT/R-B = −17.97 HU, *p* < 0.001; Δ Apex/R-CHOP = −11.2 HU, Δ Apex/R-B = −4.74 HU, *p* < 0.001; Δ anterior sulcus/R-CHOP = −10.83 HU, Δ anterior sulcus/R-B = −9.67 HU, *p* = 0.016; Δ posterior sulcus/R-CHOP = −14.69 HU, Δ posterior sulcus/R-B = −8.39 HU, *p* < 0.001).

In addition, there was a decrease in EF in the R-CHOP group (Δ EF/R-CHOP: −7.82%, Δ EF/R-B: −3.71%, *p* < 0.001).

The data therefore show an increase in epicardial fat (and thus a decrease in density), an increase in ECV and a decrease in EF (Figure 3 and Figure 4).

Considering EAT is part of visceral fat, a correlation test between EAT (mean EAT and that measured at the level of the cardiac apex and the anterior and posterior interventricular sulcus) and BMI was also conducted using pre- and post-treatment data. The correlation analysis shows a negative correlation between the radiomic variables of epicardial fat and BMI since as BMI increases, EAT decreases (Table 4).

Spearman’s correlation analysis results indicate that BMI was significantly and negatively correlated with radiological variables before and after treatment (*p* value < 0.001, Table 4). Cardiovascular events occurred in 5% of the study population. Three patients in the R-CHOP group recorded cardiological events (one patient episode of atrial fibrillation and two cases of heart failure); in contrast, one event of arterial thromboembolism was reported in the R-B group (Table 5). The analysis of PFS stratified by therapy does not show a significant difference between the two groups (*p* = 0.42) which never reaches the median (after 60 months of observation, 96.8% of patients with R-CHOP therapy and 93.2% of patients with R-B therapy did not relapse (*p* = 0.42) (Figure 5).

## 4. Discussion

FL is a generally indolent disease and although it most often responsive to treatment, it tends to be incurable and characterized by multiple recurrences in the same patient or even transformation into DLBCL [4,34,35]. New therapeutic strategies and a deeper understanding of molecular biology and the microenvironment will probably lead to a change in the natural history of the disease [34]. In view, especially over recent decades, of an improvement in PFS and OS in this patient setting, thus resulting in prolonged survival, the interest arose in identifying the best treatment, not only in terms of recovery, but also in terms of better quality of life [6,7,8]. As these patients tend to be long-term survivors, they face more long-term treatment toxicities, such as cardiotoxicity, than in the past [36,37]. As a first-line therapeutic strategy in FL patients in need of treatment, R-CHOP and R-B have proven to be effective, and studies in the literature have not demonstrated a superiority of either treatment, leading to increasing use of R-B as first-line therapy for treatment of FL [11,38].

Both regimens have side effects and, in particular, it is necessary to take into consideration the greater cardiotoxicity of the anthracyclines contained in the R-CHOP-like regimens and the greater lymphopenia and exposure to infectious events that are more commonly reported with the R-B regimen [11,38].

Our study stems from the hypothesis that early identification and application of radiological patterns can detect anthracycline-induced cardiological damage early and identify patients with occult cardiological disease onset who would benefit from anthracycline-free treatment. Studies which have investigated this topic and have used MRI or CT for the extrapolation of early markers of anthracycline-induced cardiac damage such as ECV and EAT are few in the literature and mainly concern oncology in the context of breast cancer [16,18,38,39,40,41,42].

To our knowledge, our study is the first to compare radiomic parameters such as ECV and EAT in two cohorts of FL patients treated with two different immunochemotherapy regimens.

The adipose tissue of the heart can be divided into two distinct compartments: pericardial fat and epicardial fat. The pericardial fat is outside of the parietal pericardium, while the epicardial fat is between the visceral pericardium and the myocardium [43,44,45]. Therefore, epicardial fat is a visceral fat and in adults, it is more likely to be found in the free wall of the right ventricle and in the atrioventricular and interventricular grooves [46,47]. The biochemical mechanisms are not yet fully understood. However, it appears to be a metabolically active tissue that secretes adipokines and plays an important role in the storage of free fatty acids [45,48]. However, dysregulation of and increased EAT, (whose density shifts toward lower values, being a negative number expressed in HU) may play an important role in the development of cardiovascular disease through the pro-atherogenic, pro-thrombotic and pro-inflammatory effects caused by increased epicardial fat [26,27,28,29,30,49]. In common clinical practice, patients with FL undergo a PET/CT scan at disease onset and at the end of treatment. Thus, during CT examinations, it is easy to assess adipose tissue density (EAD) [46].

Several studies in the literature have investigated the possible role of adipose tissue in inflammation by finding a correlation between EAT density and calcium in coronary arteries, increasing cardiovascular risk and making plaques vulnerable [50,51,52].

In the study by Thanassoulis G et al., an association was found between increased epicardial adipose tissue and changes in cardiac structures, such as hypertrophy of the left atrium or increased episodes of atrial fibrillation [53].

In addition, higher epicardial adipose tissue density is also associated with a higher body BMI and patients with higher visceral fat showed higher EAT [54].

In our study, only seven patients had a BMI > 30 and all of them had a pre-therapy EAT greater than −95 HU.

Monti et al. retrospectively evaluated 33 breast cancer patients and 32 controls, demonstrating a decrease in EAT density and thus an increase in visceral fat after treatment with anthracyclines [46].

Likewise, in this study, we observed a decrease in EAT in the group of patients treated with the R-CHOP regimen compared to the R-B group.

In the R-CHOP group, the use of anthracyclines can lead to cardiac damage, with necrosis of myocytes and subsequent cardiac fibrosis and thus a lower cardiac metabolism, resulting in lower EAT. Patients treated with R-CHOP are also given high doses of steroids which can cause an increase in visceral fat and lead to metabolic syndrome, which is also a risk factor for cardiovascular disease [55]. Lucijanic et al., however, demonstrated a loss of visceral and subcutaneous adipose tissue during high-dose chemotherapy in patients with DLBCL [56]. The primary aims of our study included assessing the ECV pre- and post-therapy in the two different patient groups and evaluating the ECV variation in patients treated with anthracyclines (R-CHOP group) compared to patients treated with an anthracycline-free regimen (R-B group).

Many pathological mechanisms of the heart muscle can have myocardial fibrosis as a common outcome.

It can be limited, and thus be focal fibrosis caused, for example, by myocardial cell death, or the expansion of collagen fibers in the interstitium can cause diffuse fibrosis [57].

Diffuse fibrosis can be assessed by calculating the ECV, an imaging biomarker that reflects the percentage of myocardium not composed of cardiomyocytes [18]. Its increase is indicative of a risk factor for heart failure or heart-related death [58].

Increased myocardial ECV correlates with myocardial fibrosis, cardiac amyloid or edema, which are associated with increased mortality. Quantification of ECV allows for the detection of diffuse myocardial fibrosis, which might otherwise be missed by conventional imaging [57].

Studies have shown that portal and delayed phase ECV values increase compared to pre-chemotherapy values in patients undergoing cardiotoxic chemotherapies. Notably, changes in ECV occur significantly earlier than changes observed in left ventricular ejection fraction (LVEF), which supports the notion that CT-derived ECV could be a valuable imaging marker to detect early myocardial damage and prevent cardiotoxicity in patients undergoing cancer therapy [18,59].

In this regard, we can also consider some advantages of CT at baseline for ECV over transthoracic echocardiogram (TTE).

In our study, one of the aims was to assess extracellular volume, which is not a comparable value with TTE because the biomarkers are different. In addition, we did not perform any additional CT scans for the evaluation of the disease beyond those in common clinical practice. Therefore, the assessment of ECV at baseline could in future make it possible to identify patients with a higher risk of developing cardiac damage after therapy at an early stage and thus to identify in patients who already have, for example, an increased fibrosis index (i.e., a higher ECV) those who could benefit from treatment without anthracyclines. In this way, it might be possible to identify a group of patients misdiagnosed on echocardiographic examination [60].

There are no studies investigating ECV in patients with FL, and the studies conducted are mainly in the oncology field.

Chunrong Tu, evaluating 1151 breast cancer patients, demonstrated a correlation between increased ECV and cardiological damage [59].

In our study, ECV increased statistically significantly in patients treated with R-CHOP compared to the R-B group (12.9% vs. 4.7%).

As a secondary aim, we also assessed pre- and post-therapy EF, showing a decrease in EF in patients treated with the anthracycline-containing regimen.

Our study provides initial evidence of changes in certain cardiac radiomic parameters in R-CHOP-treated patients compared to R-B-treated patients which, if confirmed, can be incorporated into the evaluation of the FL patient.

However, this study has some limitations. The first is related to its retrospective nature combined with the monocentric approach that may have led to selection bias. Hence, future prospective studies would be needed and should be extended to a larger population, including patients with lymphoma and with a longer follow-up. Monitoring these markers of possible cardiological damage over time would be interesting. The ejection fraction, as assessed by echocardiography, has the limitation of being operator-dependent and reflects a global and not a regional function.

Another limitation considered here is the low number of events reported in the study. Certainly, a larger number of events will be necessary to consolidate our conclusions regarding PFS.

The cardiological events are few (5%) and it is difficult to correlate them with the therapies carried out, considering that the elderly population is certainly affected by greater comorbidities compared to a young population.

Our study does not aim to demonstrate a superiority in either efficacy or safety between two valid and overall well-tolerated regimens, neither of which are free from side effects. Rather, our study aims to provide an analysis of radiomic changes in FL patients treated with immunochemotherapy and never before investigated in the hematological field.

## 5. Conclusions

Our study showed that R-CHOP (like) treatment in patients with FL is associated with a decrease in EAT density, an increase in epicardial fat, an increase in ECV and a decrease in EF. Thus, ECV and EAT derived from CT could be valuable imaging markers for the detection of myocardial damage and the prevention of cardiotoxicity in patients undergoing cancer therapy. This is important in patients with FL, who tend to be long-term survivors in whom short- and long-term toxicities may alter quality of life.

It becomes important to identify, especially among elderly patients with more comorbidities, such as the cohort of patients in our study, those whom we could nominate for R-CHOP-like regimens or those to be selected to be treated with equally effective anthracycline-free (R-B) regimens.

## Figures and Tables

**Figure 1 cancers-16-00563-f001:**
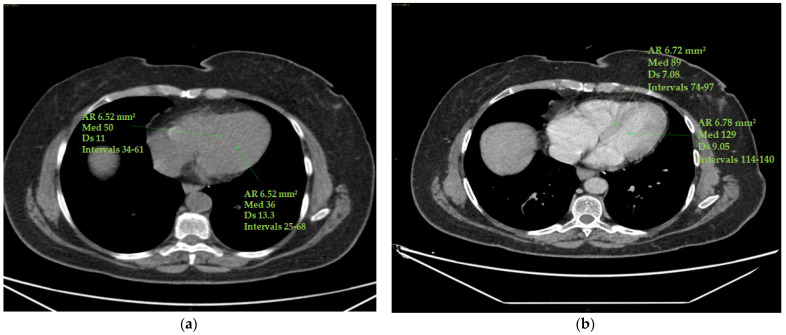
ECV measurement. Myocardial and blood pool HU values were obtained on the four cardiac chambers’ projection from an axial perspective on CT scans by manually inserting one ROI (with a mean area of 6 mm^2^ ca) in the middle of the interventricular septum and a second ROI (with a mean area of 6 mm^2^ ca) in the left ventricle’s ‘blood’ pool. Papillary muscles were avoided. The ROIs were set to the same level both in the pre-contrast scans (**a**) and in the post-contrast scans (at portal phase) (**b**). ROI standard deviations were calculated to avoid contamination of myocardial HU measurements by motion artefacts. The figure shows an example of the area of the region of interest (Ar ROI), the mean (med), the standard deviation (Ds) and the range(intervals) of hounsfield units measured.

**Figure 2 cancers-16-00563-f002:**
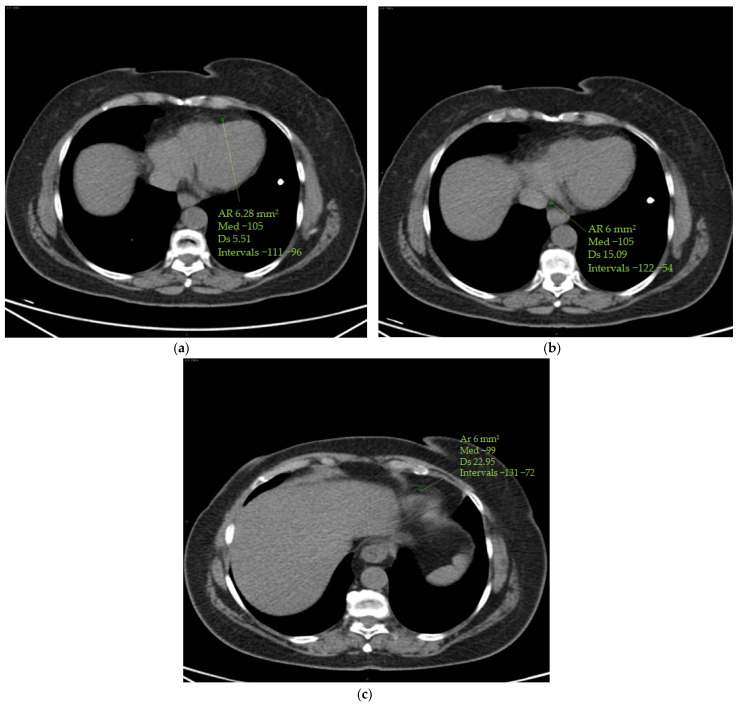
(**a**–**c**) Epicardial fat density measurement. Using a 4-chamber projection, epicardial fat density was assessed on axial perspective basal CT scans, using a manually placed ROI (with a mean area of 6 mm^2^) within the anterior interventricular sulcus (**a**), posterior interventricular sulcus (origin of the posterior interventricular artery) (**b**) and cardiac apex (**c**). The figure shows an example of the area of the region of interest (Ar ROI), the mean (med), the standard deviation (Ds) and the range (intervals) of hounsfield units measured.

**Figure 3 cancers-16-00563-f003:**
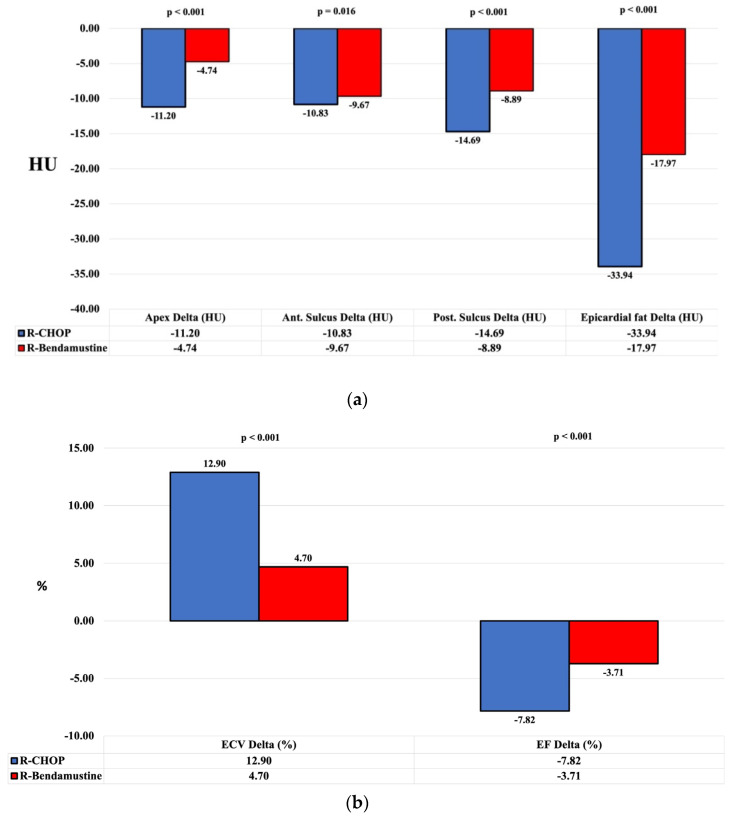
Graphical representation of “delta” mean values stratified by R-CHOP vs. R-B. Pre- and post-treatment evaluation and the change in the parameters of EAT calculated at the level of the cardiac apex and the anterior and posterior interventricular sulcus and mean EAT (**a**), EF and ECV (**b**). In the R-CHOP group, there was a decrease in the density of EAT, an increase in ECV and a decrease in EF (*p*-value < 0.001). HU: Hounsfield units; ECV: myocardial extracellular volume; EF: ejection fraction.

**Figure 4 cancers-16-00563-f004:**
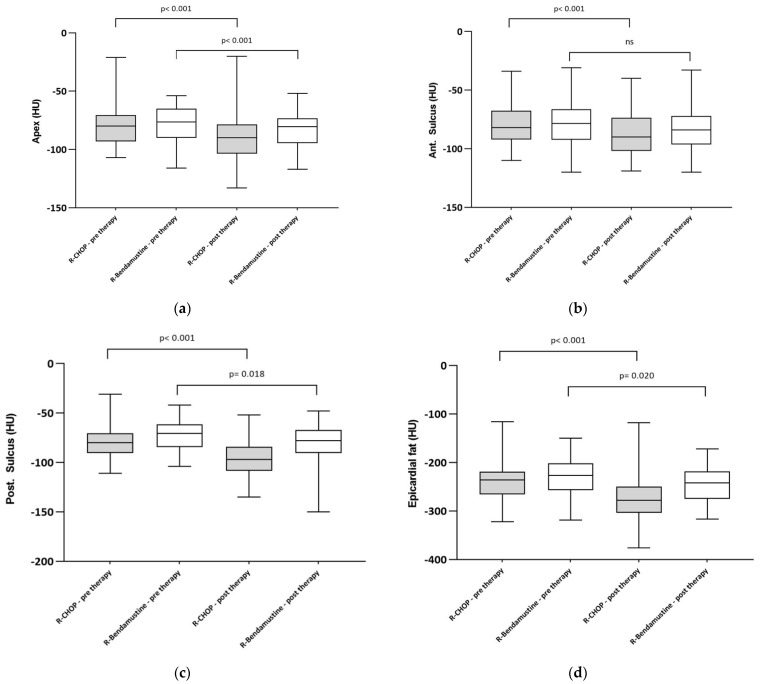
Boxplot of EAT at the level of apex (**a**), anterior intraventricular sulcus (**b**) and posterior intraventricular sulcus (**c**). (**d**) Pre- and post-treatment epicardial adipose tissue in the 2 cohorts of patients treated with R-CHOP or R-B. There was a decrease in EAT density (and thus an increase in adipose tissue) in the R-CHOP group. Boxplots show median and interquartile range. Statistical significance between groups is shown.

**Figure 5 cancers-16-00563-f005:**
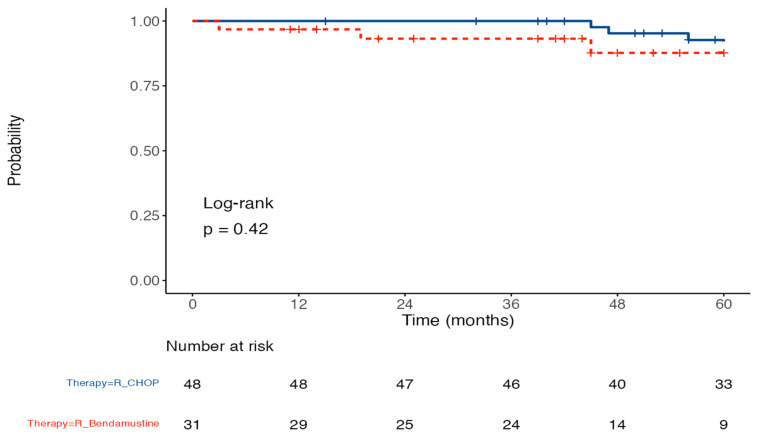
PFS probability of the patients treated with R-CHOP and R-B. Kaplan–Meier curves of progression-free survival in the R-CHOP and R-B groups (*p* value = 0.42).

**Table 1 cancers-16-00563-t001:** Presentation characteristics stratified by R-CHOP vs. R-bendamustine therapy.

Population (*n* = 80)	R-CHOP (*n* = 49)	R-Bendamustine (*n* = 31)	*p*-Value
Median age, years (range)	66 (52–77)	70 (59–81)	0.053
Male/Female ratio	1.0 (25:24)	0.7 (13:18)	0.428
Median height, cm ± SD (range)	166.2 ± 9.1 (150.0–189.0)	167.5 ± 7.9 (150.0–184.0)	0.498
Median weight, kg ± SD (range)	75.8 ± 21.2 (48.0–187.0)	72.0 ± 14.2 (48.0–97.0)	0.377
Median BMI ± SD (range)	22.9 ± 6.3 (16.0–56.7)	21.4 ± 3.8 (14.8–28.8)	0.227
Grading			
G1	4.2% (2/48)	16.1% (5/31)	0.241
G1–G2	16.7% (8/48)	16.1% (5/31)	
G2	47.9% (23/48)	42.0% (13/31)	
G2–G3A	6.2% (3/48)	12.9% (4/31)	
G3A	25.0% (12/48)	12.9% (4/31)	
FLIPI s			
Low	2.1% (1/48)	9.7% (3/31)	0.322
Intermediate	58.3% (28/48)	54.8% (17/31)	
High	39.6% (19/48)	35.5% (11/31)	

SD: standard deviation; BMI: body mass index; FLIPI: Follicular Lymphoma International Prognostic Index. Mann–Whitney test; Student’s *t*-test; Wilcoxon signed-rank tests.

**Table 2 cancers-16-00563-t002:** Mean (±SD) radiomic pre- and post-therapy values in the R-CHOP and R-B group.

Population (*n* = 80)	R-CHOP			R-Bendamustine		
Primary Endpoint	Pre-Therapy	Post-Therapy	*p*-Value	Pre-Therapy	Post-Therapy	*p*-Value
Mean ± SD Apex (HU)	−79.12 ± 16.85	−90.33 ± 19.27	<0.001	−78.55 ± 15.09	−83.29 ± 14.66	<0.001
Mean ± SD Anterior Sulcus (HU)	−79.12 ± 16.88	−87.16 ± 18.15	<0.001	−76.64 ± 18.24	−81.48 ± 19.05	0.139
Mean ± SD Posterior Sulcus (HU)	−79.84 ± 18.54	−94.53 ± 20.23	<0.001	−72.39 ± 15.04	−80.77 ± 20.28	0.018
Mean ± SD Epicardial Fat (HU)	−238.08 ± 43.97	−272.02 ± 49.93	<0.001	−227.58 ± 41.17	−245.55 ± 38.52	0.020
Mean ± SD ECV (%)	0.304 ± 0.060	0.432 ± 0.097	<0.001	0.315 ± 0.063	0.361 ± 0.117	0.022
Mean ± SD EF (%)	64.2 ± 3.6	56.4 ± 16.4	0.001	63.4 ± 3.3	61.7 ± 3.8	0.013

SD: standard deviation; HU: Hounsfield units; EF: ejection fraction; ECV: myocardial extracellular volume.

**Table 3 cancers-16-00563-t003:** Comparison of radiomics delta means values results of R-CHOP vs. R-bendamustine therapy.

Population (*n* = 80)	Mean ± SD	95% CI	Mean ± SD	95% CI	*p*-Value
Apex Delta (HU)	−11.20 ± 8.72	(−13.70–−8.70)	−4.74 ± 9.07	(−8.07–−1.41)	<0.001
Apex Delta (%)	−14.67 ± 13.66	(−18.59–−10.75)	−7.10 ± 14.12	(−12.29–−1.92)	<0.001
Anterior Sulcus Delta (HU)	−10.83 ± 11.77	(−14.21–−7.44)	−9.67 ± 28.31	(−20.26–0.51)	0.016
Anterior Sulcus Delta (%)	−8.04 ± 8.34	(−10.44–−5.64)	−4.84 ± 17.86	(−11.39–1.71)	0.016
Posterior Sulcus Delta (HU)	−14.69 ± 10.90	(−17.83–−11.56)	−8.39 ± 18.72	(−15.25–−1.52)	<0.001
Posterior Sulcus Delta (%)	−20.96 ± 22.53	(−27.43–−14.49)	−13.64 ± 30.55	(−24.85–−2.44)	<0.001
EAT Delta (HU)	−33.94 ± 20.09	(−39.71–−28.17)	−17.97 ± 40.44	(−32.80–−3.13)	<0.001
EAT Delta (%)	−14.61 ± 9.98	(−17.47–−11.74)	−10.17 ± 22.62	(−18.47–−1.88)	<0.001
ECV (%) Delta	12.9 ± 8.8	(10.3–15.4)	4.7 ± 10.8	(0.8–8.7)	<0.001
ECV Delta (%)	−45.42 ± 37.41	(−56.28–−34.55)	−16.18 ± 35.96	(−29.37–−2.99)	<0.001
EF (%) Delta	−7.82 ± 16.01	(−12.41–−3.22)	−3.71 ± 11.87	(−8.06–0.64)	<0.001
EF Delta (%)	12.21 ± 26.61	(4.57–19.85)	5.67 ± 18.23	(−1.01–12.36)	<0.001

Abbreviations: SD: standard deviation; 95% CI: 95% confidence interval; ECV: myocardial extracellular volume; EF: ejection fraction; HU: Hounsfield units; EAT: epicardial adipose tissue.

**Table 4 cancers-16-00563-t004:** Spearman’s correlation test (Rho) to analyze the relationship between BMI and pre- and post-therapy cardiac radiomic variables.

Variable	Rho	*p*-Value
Apex pre-therapy (HU)	−0.538	<0.001
Apex post-therapy (HU)	−0.463	<0.001
Ant. sulcus pre-therapy (HU)	−0.367	<0.001
Ant. sulcus post-therapy (HU)	−0.294	0.009
Post. sulcus pre-therapy (HU)	−0.429	<0.001
Post. sulcus post-therapy (HU)	−0.482	<0.001
Epicardial fat post-therapy (HU)	−0.500	<0.001
Epicardial fat post-therapy (HU)	−0.507	<0.001

Abbreviation: HU: Hounsfield units; Spearman’s correlation test (Rho).

**Table 5 cancers-16-00563-t005:** Patients with cardiological AES (5%) in the R-CHOP and R-B groups.

Patient No.	Gender	CTCAE	Therapy	BMI	Apex Delta (HU)	Ant. Sulcus Delta (HU)	Post. Sulcus Delta (HU)	EF Delta (%)	ECV Delta	EF Delta (%)
1	F	Atrial fibrillation (AF)	R-CHOP	22.8	−12	−23	−21	−56	0.21	−13
2	F	Heart failure	R-CHOP	21.8	−9	−10	−14	−33	0.15	−6
3	M	Heart failure	R-CHOP	22.5	−10	−12	−10	−32	0.22	−12
4	F	Arterial thromboembolism	R-bendamustine	20.9	0	−2	−2	−4	0.10	0

CTCAE: common terminology criteria for adverse events; BMI: body mass index; ECV: myocardial extracellular volume; HU: Hounsfield unit; EF: ejection fraction.

## Data Availability

The data presented in this study are available on request from the corresponding author.

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
