# Peer review of "CT Images in Follicular Lymphoma: Changes after Treatment Are Predictive of Cardiac Toxicity in Patients Treated with Anthracycline-Based or R-B Regimens"

_cancers, 2024, doi:10.3390/cancers16030563_

Round 1

Reviewer 1 Report

Comments and Suggestions for Authors

Interesting correlational study.

Please correct the english/grammar/spelling.

For example

Key words: extracellular miocardial volum vs extracellular mYOCARDIAL 

Line 46 Follicular lymphoma is a generally indolent disease and vs

Follicular lymphoma is generally an indolent disease and 

Line 51: with different prognoses based vs with VARIABLE PROGNOSIS based

Line 113: n the future, these parameters could help us in therapeutic choice and in the evalu-113 ation of a possible prognostic value.   PLEASE REVISE.

Line 438  in elderly patients, such as our study population)with. PLEASE REVISE.

Please have the paper reviewed by an english expert before publication.

Comments on the Quality of English Language

Interesting correlational study.

Please correct the english/grammar/spelling.

For example

Key words: extracellular miocardial volum vs extracellular mYOCARDIAL 

Line 46 Follicular lymphoma is a generally indolent disease and vs

Follicular lymphoma is generally an indolent disease and 

Line 51: with different prognoses based vs with VARIABLE PROGNOSIS based

Line 113: n the future, these parameters could help us in therapeutic choice and in the evalu-113 ation of a possible prognostic value.   PLEASE REVISE.

Line 438  in elderly patients, such as our study population)with. PLEASE REVISE.

Please have the paper reviewed by an english expert.

Author Response

We acnoledge receipt of your  comments which have been object of all our attention and we thank you. In the following we report all the changes in accordance with your suggestions  which are reported as well in the manuscript.

According to your general english editing comments, please be aware that we have provided the manuscript revisions by an english expert and the update is reported in the manuscript test.

Answers to your questions attached 

Reviewer 2 Report

Comments and Suggestions for Authors

Thank you for the authors for this paper. However, there are some corrections/upgradings/clarifications needed:

-The simple summary contains quite many times "We did something". Usually the scientific language is in passive and shorter and more precise. Please sharpen this paragraph.

-Line 57: please remove the word unexplained, as the lymphoma explains the cytopenias. Or change to cytopenias that are not otherwise explained.

-Line 57: The natural history of FL has not changed in recent history but the prognosis is. Please correct. The same issue is in the line 333.

-Line 60: The OS is improved only in some patients, not all and the wording should be clarified.

-Line 63: R-COMP is not frequently used and should not be mentioned here.

-Line 70: the grade 3B in FL is not a sign of transformation to DLBCL. This grade behaves like DLBCL and it is a different thing.

-Line 71: Actually there are another thing that prefer the use of R-CHOP: if patient need a rapid response to treatment. And R-benda is preferred, if patient has previous cardiac failure.

-Line 82: The older patients more often receive R-CVP than R-COMP. Please clarify.

-Line 82 (+line 108): Cardiac troponine is considered as an early marker of cardiac damage, although it is not specific.

-Line 89: Range cannot be a one figure. Please clarify. 

-Line 102: Please describe, what is a low-density EAT.

-Line 118: Since May 2019 there has not been yet 60 months, so please clarify the minimum follow-up time.

-Line 120: 120 patients minus 41 patients, but still 80 patients were in the analysis (line 125). Please clarify.

-Line 209: Please clarify the mentioned sample size 78 (previously 80).

-Table 2: the table text says that here is comparison of results between different therapies. However, it seems that p-values are counted between pre- and post-therapy in one treatment group and there is no comparison between the groups. Please clarify.

-The Results paragraph should be rethought to be more clear. 

-Please re-name the subfigures of figure 4 to a-d and modify the figure text accordingly.

-What is the benefit of baseline CT for ECV compared to transthoracic echocardiogram with GLS measurement which is usually normally done?

-Have you compared your results with echocardiography? How similar the results of EF are?

-The R-COMP should be safer to patients regarding cardiac events. Any difference between those and R-CHOP patients?

-Any differences when counted according to BMI groups in these parametres?

Comments on the Quality of English Language

-Moderate language revising is needed, spaces missing, sometimes there is not capital letters when needed and sometimes they are, when not needed. Paragraphs are short and thus the text is not fluent.

-Line 51: lymphoma should not be written in capital letter.

-Line 244: Please correct: 2.1% in R-CHOP).

-Table 2: Please correct FE to EF and list it in the abbreviations below. Please use the same abbreviation through the whole text (be consistent). Also in the figure 3 and other tables there is FE instead of EF.

-Line 277: Please correct to p-value <0.001, table 4).

Author Response

We acknoledge receipt of your  comments which have been object of all our attention and we thank you.

In the following we report all the changes in accordante with your suggestions. and  reported as well in the manuscript.

Reviewer 3 Report

Comments and Suggestions for Authors

I have truly enjoyed reading the paper investigating dynamics of CT characteristics considering cardiotoxicity in FL patients treated with two immunochemotherapy regimens. 

The paper is overall well written and well presented. I have only minor comments as follows:

1) Authors should recognize in the limitations section of the discussion that in the real life different patients are candidates for CHOP and Bendamustin regimens, mostly based on comorbidities and age and those treated with CHOP might have higher cardiotoxicity and those treated with bendamustine might be more prone to infectious complications and might have higher non-relapse mortality as shown in some previous studies.

2) The number of events in the current study was low, resulting in insufficient statistical power to provide conclusions regarding PFS but authors took decent effort to provide such analysis considering two different regimens. Please recognize this among limitations of the paper.

3) What was the contribution of COVID-19 infection on observed parameters if data are available? 

4) The paper would benefit from expanding the discussion to comment on findings reported in DLBCL patients treated with CHOP like regimen regarding dynamics of CT assessed visceral adipose tissue over time (doi: 10.1080/10428194.2022.2034160). 51% of DLBCL patients seem to experience visceral fat tissue loss (assessed as perirenal fat measurements) and 41% loss of subcutaneous fat during chemotherapy. More pronounced fat tissue loss was not associated with better or worse response to therapy. However, more pronounced fat tissue lass was associated with worse prognosis regarding OS and PFS, however, loss of visceral fat did not affect prognosis as much as loss of subcutaneous fat. 

5) last sentence of the discussion is unclear (Rather, our study aims to be a first analysis of radiomic evaluations explored in the oncology and never hematology field.). I suggest to rephrase and finish the sentence with "Rather, our study aims to provide analysis of radiomic changes in FL patients treated with immunochemotherapy."

Author Response

We acknoledge receipt of your  comments which have been object of all our attention and we thank you.

In the following we report all the changes in accordante with your suggestions  reported as well in the manuscript.

Regards

Dr Fabiana Esposito

Round 2

Reviewer 2 Report

Comments and Suggestions for Authors

Thank you for corrections. However, there are still some points to be corrected.

-Line 72: there is still said "there is no other indication of preference", but the next paragraph says the rapid response and heart failure. These are contradictionary to each other and the former sentence should be corrected.

-The title of Table 2 should be corrected to clarify what is done. It is not now in line with the given numbers.

-The results paragraph: Lines 259-261 are still unclear. Also lines 290-295 are not understandable as it has 6 lines of different numbers and symbols, without any explanation what these mean (to a table?, only explanation, what changed?) Line 299: what was the result? If it is said in the next paragraphs, there should be a separated paragraph from the previous sentence. 

-The benefit of baseline CT for ECV compared to TTE should be clarified also to the text.

Comments on the Quality of English Language

-Line 21: R-bedamustine is missing n.

-line 59: benefits has an unnecessary s. Should it be: have benefitted?

-Line 260: 54.8 is missing %.

Author Response

Answers for reviewer 2 are attached.
